# Hyperthermic Intrathoracic Chemotherapy for Malignant Pleural Mesothelioma: The Forefront of Surgery-Based Multimodality Treatment

**DOI:** 10.3390/jcm10173801

**Published:** 2021-08-25

**Authors:** Vittorio Aprile, Alessandra Lenzini, Filippo Lococo, Diana Bacchin, Stylianos Korasidis, Maria Giovanna Mastromarino, Giovanni Guglielmi, Gerardo Palmiero, Marcello Carlo Ambrogi, Marco Lucchi

**Affiliations:** 1Unit of Thoracic Surgery, Department of Critical Area and Surgical, Medical and Molecular Pathology, University of Pisa, 56122 Pisa, Italy; vittorio.aprile@unipi.it (V.A.); alessandralenzini2@gmail.com (A.L.); stylianoskorasidis@gmail.com (S.K.); mgmastromarino@gmail.com (M.G.M.); m.ambrogi@med.unipi.it (M.C.A.); marco.lucchi@unipi.it (M.L.); 2Thoracic Surgery Unit, Fondazione Policlinico Universitario A. Gemelli IRCCS, 00168 Rome, Italy; filippo_lococo@yahoo.it; 3Occupational Health Department, U.O. Medicina Preventiva del Lavoro, Azienda Ospedaliero-Universitaria Pisana, 56122 Pisa, Italy; G.guglielmi@ao-pisa.toscana.it; 4Pneumology Unit, Versilia Hospital, 55049 Camaiore, Italy; rockinfreak30@gmail.com

**Keywords:** mesothelioma, local treatment, HITHOC, hypertermia, chemotherapy

## Abstract

Introduction: Malignant Pleural Mesothelioma (MPM) is characterized by an aggressive behavior and an inevitably fatal prognosis, whose treatment is still far from being standardized. The role of surgery is questionable since a radical resection is unattainable in most cases. Hyperthermic IntraTHOracic Chemotherapy (HITHOC) combines the advantages of antitumoral effects together with those of high temperature on the exposed tissues with the aim to improve surgical radicality. Material and Methods: this is a narrative review on the role of HITHOC in the management of MPM patients. To provide data on the beginnings and the historical evolution of this technique, we searched the available literature by selecting the more exhaustive papers on this topic. Results: from 1994 to date different authors experimented HITHOC following a cytoreductive surgery in MPM, obtaining in most cases a good local control and a better overall survival associated to very low complication rate. Conclusions: HITHOC may be considered as a safe, feasible and effective procedure although there is a high heterogeneity between different protocols adopted worldwide. More structured studies are needed to reach a unanimous consensus on this technique.

## 1. Introduction

Malignant Pleural Mesothelioma (MPM) is a rare neoplasia arising from mesothelial cells lining serous cavities, characterized by an aggressive behavior and an inevitably fatal prognosis.

Asbestos exposure is universally recognized as the main risk factor of developing MPM as class 1 carcinogenic agents by the International Agency for Research on Cancer (IARC), although after a latency period that may be very long (up to 50 years). Therefore, its treatment is of great topicality since incidence peak is expected just during the next years and will last for long time before this trend reverses [1].

Early symptoms of MPM are generally non-specific as most patients present breathlessness and chest pain, but in certain ones may occur fatigue, weight loss and sweating; consequently, most patients affected were addressed to specialists in advanced disease whereas prognosis is generally poor with median survival ranging from 8 to 14 months [2].

Histology and pathological stage deeply affect long-term outcomes, and World Health Organization (WHO) recognized three main histological subtypes: epithelioid, sarcomatoid and biphasic, each one with specific pathological features and prognosis. MPM with pure epithelioid histology (50–60% of cases) is generally associated to a significant prolonged survival compared to biphasic and sarcomatoid histology (14.4 months vs. 5.4–9.3 months) [3,4].

Due to MPM features and the typically advanced age of patients at the diagnosis, treatment options are extremely limited; where surgical role remains controversial and reserved to selected patients after a multidisciplinary tumor board assessment; while most patients were addressed to chemotherapy with platinum plus pemetrexed in first line as standard of care or to radiation-therapy, this latter mainly with palliative intent or in a adjuvant setting [5].

To date, because of the scarce success of any single modality of treatment in affecting long-term outcomes, the prognosis of MPM affected patients remains extremely poor although many hopes hang on other novel strategies based mainly on immunotherapy [6].

Nevertheless, in the last decades, a variety of multimodality approaches, including the surgery-based ones, have been proposed to treat MPM. Surgery, in fact, plays an important role in all steps of management of MPM at any stages, from diagnosis to staging, treatment and, lastly, palliation.

Surgery assumed a curative role from ’70s, when Butchart and colleagues introduced pleuropneumonectomy to treat selected patients affected by MPM [7].

Since then, different surgical procedures with curative intent have been described that could be summarized in:Extrapleural Pleuropneumonectomy (EPP) described as “en-bloc” resection of parietal and visceral pleura, the homolateral lung, the pericardium and the diaphragm in order to achieve a resection as much radical as possible.Extended Pleurectomy and Decortication (EPD): defined as the resection of both parietal and visceral pleura together with pericardium and hemidiaphragm [8].Pleurectomy/Decortication (P/D) consisting in the resection of purely both pleurae.

Different studies and two randomized trials tried to demonstrate the effectiveness of each treatment over the others, unfortunately with no clear answers, although there is a tendency towards a less demolitive surgery inasmuch more aggressive procedures significantly impair quality of life and reduce the chance of further treatments [9,10,11].

In MPM, microscopic complete resection (R0) represents an unattainable goal; pleura anatomy itself and the laminar tumor growth, together with the aggressive behavior of the tumor, that spreads early to the surrounding structures, drastically reduce the possibility of achieving disease-free margins and several authors, proposed to set as surgical target a Macroscopic Complete Resection (R1) [12].

Potential tumoral residues has led to test intracavitary therapies as Hyperthermic Intrathoracic Chemotherapy (HITHOC) to improve loco-regional effect of surgery despite lower impact in terms of systemic toxicity [13].

## 2. Materials and Methods

Relevant literature up to January 2021 was searched in PubMed using as keywords: “hyperthermic intrapleural chemotherapy”, “intrapleural hyperthermic” or “hyperthermic intrathoracic chemotherapy”, or “HIPEC” and “mesothelioma”, or “HITHOC”. The search was limited to English language and relevant studies were identified, screened and reviewed by all the authors.

We conducted an accurate research of all studies focused on the outcomes of HITHOC in MPM since 1994, and we selected only those with information about HITHOC protocol proposed and information on postoperative course as well as oncological outcomes. Unpublished material, congress abstracts and proceedings were not considered.

## 3. Hyperthermic Intracavitary Chemotherapy

HITHOC is part of surgery-based multimodality therapy proposed for primary or secondary serous surfaces malignancies to improve tumor local control and, consequently, prognosis. This technique combines the advantages of antitumoral effects together with those of high temperature on the exposed tissues. The underlying rationale is that locally administered chemotherapy may enhance the efficacy of chemotherapeutic agents by achieving high levels of drugs with limited systemic toxicity, while hyperthermia improves the efficacy and penetration of chemotherapy (Table 1).

Since ’80s, many studies demonstrated that a mild hyperthermia (40–42 °C) has a direct cytotoxic effect and improves local effects of chemotherapeutic agents. Intracavitary thermochemotherapy was firstly applied to peritoneal carcinomatosis under the acronym HIPEC, coined by the Netherlands Cancer Institute and it became the standardized nomenclature for this procedure [14].

Direct intra-peritoneal instillation of anticancer agents was firstly suggested by Dedrick and Markman for ovarian tumors in the late 1960 [15]. Since then, HIPEC is generally performed after surgical debulking of advanced abdominal cancers, after removing as much peritoneal tumoral areas as possible.

The first intraperitoneal application of hyperthermia by perfusion, indeed, was successfully performed in the 1980 by Spratt et al. [16] for a case of pseudomyxoma peritonei; since that time, a growing number of centers worldwide reported the use of this technique.

In vivo malignant cells, in fact, appear to be selectively destroyed by hyperthermia in the range of 41–43 °C, because of various mechanisms as nucleic acids and protein synthesis alterations, mitotic arrest, depression of aerobic tumor cells metabolism and an increased lysosomal activity that led to cytoplasmatic damage [17]. Moreover, immune system response can be activated against tumor cells by moderate hyperthermia, leading to the activation of heat shock proteins and releasing of exosomes that stimulate natural killer and CD8+ T cells [18].

Many chemotherapeutic agents, moreover, show synergism with hyperthermia: the synergism is predominantly dependent on the increased drug uptake especially in malignant cells, due to increased membrane permeability and improved membrane transport.

The choice of the chemotherapeutic agent represents a crucial aspect: an ideal drug should have a well-defined activity against the malignancy with minimal or absent local and systemic toxicity after administration. As a matter of fact, many chemotherapeutic agents suitable for HITHOC have shown to be related to various collateral effects. Among them, acute kidney injury (AKI)m is the most frequent, especially when cisplatin-based regimens are used, followed by bone narrow toxicity and gastrointestinal toxicity. In this regard Hod et al. recently conducted an observational study to assess the risks of AKI after cytoreductive surgery followed by cisplatin-based HITHOC in 512 patients. They found out that approximately half of the patients developed a mild stage of AKI, while only 3.2% of patients required renal replacement therapy. Moreover, the authors identified among the significant predictors for renal failure male sex and intraoperative cisplatin administration. Nevertheless, severe AKI was observed especially in patients who received high doses of cisplatin (225 mg/m^2^); for this reason, the authors reviewed HITHOC protocol and reduced cisplatin dosage to 175 mg/m^2^ [19].

From the beginning, many authors experimented different combinations of chemotherapeutic agents with direct cytotoxic effect at various degrees of hyperthermia, trying to minimize systemic toxicity. For this purpose, pre-treatment hydration protocols in association with cytoprotective drugs administration were implemented [20].

Ratto et al. [21] demonstrated that hyperthermic intrathoracic perfusion with cisplatin had serious pharmacokinetics advantages with low systemic toxicity; in this study, 10 patients with epithelial or mixed, stage I or II, MPM underwent cytoreductive surgery followed by intrapleural normothermic or hyperthermic perfusion. They showed that the local tissue/perfusate ratio of platinum concentrations tended to be higher after hyperthermic perfusion rather than normothermic perfusion.

Rusch and colleagues evaluated, for the first time in 1994, the pharmacokinetics of intrapleural cisplatin, used in combination with mitomycin [22], reporting that systemic absorption was rapid, with significantly higher levels of chemotherapeutic in the pleural fluid than in the plasma and peak plasma levels reached within one hour from intrapleural administration.

In almost all study, Cisplatin appeared as a safe therapeutic option for intrapleural perfusion, even if there is still no consensus about the optimal dosage, with intrapleural doses described between 80 and 250 mg/m^2^.

Cisplatin was used in combination to Doxorubicin, by Van Ruth and colleagues between 1998 and 2001 in a series of 24 patients who underwent HITHOC for MPM; in this study authors evaluated the potential pharmacokinetic of Doxorubicin that showed a minimal systemic uptake and good local effect [23].

Therefore, as concerns thoraco-abdominal field, Mitomycin (15 mg/m^2^) was used by Sugarbaker et al. in combination to Doxorubicin (15 mg/m^2^) for hyperthermic chemotherapy perfusion following cytoreductive surgery and eventual diaphragm excision in 60 patients affected by peritoneal/pleural carcinosis from gastro-intestinal cancers or peritoneal mesothelioma. After 90 min of treatment the authors measured plasma drug levels and they found out that serous absorption was higher in the patients who underwent intraperitoneal chemotherapy than in those who underwent thoraco/abdominal and intrapleural perfusion. This study led to the conclusion that intrapleural administration of drugs has an acceptable safety profile [24].

Nowadays, HITHOC is performed in several referenced centers, mainly in case of MPM and pleural spread of thymoma, even if, some authors, reported its use also for unilateral pleural carcinomatosis lung cancer-related and other primitive tumors.

## 4. Perfusion Technique

A dedicated perfusion system is mandatory to perform HITHOC, with a software for loco-regional oncological treatment. This kind of system, such as the one used by Ambrogi and colleagues [25] in their experience, offers the possibility of automatic regulation, the ability to control the perfusion of the pleura space and the achievement of high temperature thanks to a dedicated heat exchanger. Perfusion volume is preloaded in the machine according to body surface, in order to have in all patients the same proportion of chemotherapy agents. The device is then connected to inflow and outflow drainages; consequentially, pleural space is gradually filled with isotonic saline solution at a temperature of 38 °C, which is progressively increased up to obtain intrathoracic 42.5 °C. At this moment, chemotherapeutic agents are added to the system and their perfusion lasts at least 60 min. In the last step, inflow and outflow catheters are exchanged in order to evacuate all the perfusion solution and temperature probes van be removed.

## 5. The Role of Hithoc on MPM

Survival rates in MPM patients are still unsatisfying mainly due to a frequently late diagnosis, impaired physical conditions of the patients and the relative ineffectiveness of each single treatment. Hence, a multimodal approach is generally recommended, even if only few patients are eligible for it, especially when systemic chemotherapy is indicated [13].

In this scenario, HITHOC following surgery emerges as a valid attempt to improve the local disease control in those patients affected by early stage MPM, whereas tumor is confined to an hemithorax and to superficial layers of the pleura, within the framework of a multidisciplinary treatment.

To date, there is no standardized protocol for HITHOC, and its clinical application has not yet been clearly defined in recent guidelines or task force by ERS/EACTS/ESTS/ESCRO. For these reasons most of literature in this topic is limited to retrospective analysis with different technique, protocol and chemotherapeutic agents, at various doses even if the majority of these studies reported promising results [26].

In Table 2 and Table 3 are reported technical details and outcomes of several studies focused on this topic.

Among the first studies who described HITHOC in the management of MPM, deserves a special mention the one published by Rusch and colleagues in 1994, who presented a phase-II trial conducted to investigate the feasibility of a combination of surgical resection and intrapleural chemotherapy and subsequent systemic chemotherapy. The study included 36 patients with resectable MPM, 28 of them underwent P/D and intrapleural chemotherapy with cisplatin 100 mg/m^2^ and mitomycin 8 mg/m^2^, followed by systemic chemotherapy. The overall survival rate was 68% at 1 year and 40% at 2 years, with a median survival duration of 17 months [27].

Yellin and colleagues in 2001 presented their experience in 26 patients (7 of them with MPM) who underwent intraoperative hyperthermic pleural perfusion with cisplatin, from 1994 to 1998. Of the 7 patients with MPM, 4, 2 and 1 patients underwent EPP, surgical exploration and pleurectomy, respectively, followed in all cases by HITHOC with Cisplatin (150–200 mg). In this study, one patient died postoperatively due to operative complications while three patients survived for more than 2 years [28].

In 2006 Richards et al. proposed for the first time a prospective study to evaluate the maximum-tolerated dose and the outcome in 61 patients treated with intraoperative intracavitary hyperthermic cisplatin washing after extended pleurectomy for MPM. In this study, authors reported significantly better survival rates in patients with epithelial tumors (19 months vs. 8 months with sarcomatoid/mixed cells form) and in those treated by high-dose cisplatin (175 to 250 mg/m^2^) lavage (18 months vs. 6 months of patients exposed to lower doses) [29].

In 2009, Zellos published a phase I study aimed to determine toxicity of intraoperative intracavitary hyperthermic cisplatin perfusion with amifostine after EPP for MPM. A series of 29 patients between August 2001 and July 2002 underwent a 1-h hyperthermic cisplatin perfusion (42 °C) of ipsilateral hemithorax and abdomen with amifostine, after surgery. Mortality rate reported was 7%, whereas median survival was 17 months, with better result in patients with resectable disease (20 months), early stage (35 months) and in whose received higher cisplatin doses up to 200 mg/m^2^ (26 months) [30].

In the same year, Tilleman and al. proposed a phase II prospective study enrolling 121 patients affected by MPM. 92 patients underwent EP followed by hyperthermic intraoperative cisplatin (225 mg/m^2^) perfusion and 27 of them received also amifostine (910 mg/m^2^). Recurrence of MPM was documented in 47 and the median survival of all patients was 12.8 months. The results showed that the use of amifostine might reduce cisplatin-associated renal toxicity [31].

Sugarbaker and colleagues in 2013 performed a prospective phase I-II trials to describe the outcome of low-risk patients with MPM who underwent cytoreductive surgery followed by intraoperative intracavitary hyperthermic cisplatin chemotherapy. Cytoreductive surgeries (74 EPP and 29 P/D) were performed for 103 low-risk patients from 2001 to 2009. In this case, 72 patients received intraoperative intracavitary hyperthermic cisplatin 175 to 225 mg/m^2^ chemotherapy for a 1-h lavage at 42 °C and 31 patients did not. The intraoperative intracavitary hyperthermic cisplatin chemotherapy group (72 patients) exhibited a significantly longer disease-free interval (27.1 vs. 12.8 months) and overall survival (35.3 vs. 22.8 months) than the comparison group (31 patients). Among the low-risk patients, treatment with intraoperative intracavitary hyperthermic cisplatin chemotherapy appeared to be particularly beneficial for patients with stage N1-N2 nodal involvement and for those who did not undergo adjuvant therapy [32].

Ambrogi et al. in 2018, proposed his experience with hyperthermic chemotherapy following diaphragm and lung–preserving surgery for malignant pleural mesothelioma; 49 patients between 2005 and 2014, with epithelioid or biphasic malignant pleural mesothelioma were treated with lung–diaphragm– pericardium-sparing pleurectomy followed by HITHOC with a combination of Cisplatin (80 mg/m^2^) and Epirubicin (25 mg/m^2^). Authors reported a median overall survival of 22 months with a statistically significant difference between early stage and advanced 35 months vs. 17 months) [25].

In the same year, Burt and colleagues performed a clinical phase I trial on 104 patients to establish the maximum tolerated dose of gemcitabine added to cisplatin during intraoperative chemotherapy after surgical resection of MPM; the study revealed a maximum tolerated dose of 175 mg/m^2^ in the case of cisplatin and 1000 mg/m^2^ in the case of gemcitabine. The secondary objective of the authors was to analyze overall survival and recurrence-free survival: the median overall survival in patients who underwent EPP and P/D were 17.7 and 38.8 months, respectively, with a better outcome for epithelioid histologic type compared to the nonepithelioid one. Furthermore, in multivariable analyses, a decreased recurrence free survival was associated to several factors as nonepithelioid histologic type and advanced pathologic stage [33].

More recently, Klotz analyzed the clinical outcome of 71 patients affected by MPM, treated with P/D followed by HITHOC with cisplatin and doxorubicin. Peri-operative morbidity appeared to be acceptable, since more than 40% of all complications were classified as minor and successfully relieved by conservative treatment. They observed a median overall survival of 16.1 months, significantly influenced by histological pattern: 9.2 months for sarcomatoid, 10.9 months, for biphasic and 17.9 months for epithelioid type. Multivariate analysis confirmed that histologic tumor subtype and radicality of resection impact overall survival rates as independent risk factors [34].

## 6. Discussion

Improvement of diagnostic and therapeutic technologies together with progress in biomolecular analysis allowed a deeper understanding of MPM; nevertheless, the optimal treatment of this disease is still an unsolved question. To date, better outcomes have been observed in patients addressed to multimodality therapy, even though there is still an open debate on which treatment regimen should be recommended. The role of surgery with curative intent is highly questionable: surgeons’ goal is achieving a complete macroscopic resection (R1) of the tumor or a maximal cytoreduction since there is no chance to reach microscopic resection (R0) due to the anatomical and biological features of this tumor.

Therefore, additional strategies, such as adjuvant chemotherapy or radiotherapy, are required to control tumor residues to increase surgery radicality; even if additional intracavitary therapies in a multimodal approach allow a less aggressive surgical resection, the surgical technique to prefer is still unclear. Whether EPP or P/D is a better surgical choice is still matter of debate.

HITHOC may be considered as a safe, feasible and effective local treatment to improve the local effect of surgery, but even if many studies show promising results HITHOC has not been discussed in the last guidelines of the task force of the ERS/EACTS/ESTS/ESCRO on treatment of MPM, as Migliore and colleagues have already noticed [26].

Zhao et al. [35] carried out a systematic review and meta-analysis of selected 21 papers and found that median survival time of the MPM patients who received HITHOC in addition to surgical resection including extrapleural pneumonectomy (EPP) and pleurectomy/decortication (P/D) was significantly prolonged. However, the attempt to analyze all the considerable experiences with HITHOC carried out in the last two decades is limited by the fact that techniques used to perform HITHOC are extremely heterogeneous including surgical aspects and differences of antitumoral drugs perfused, their dosage, perfusion machine, temperature and time of the perfusion solution.

We are still far from reaching the optimal treatment for MPM patients, which should provide a prolonged disease control without impairing extensively the quality of life.

Designing a prospective randomized study on large series of MPM patients represents a major challenge because of the rarity of the tumor itself and its high mortality rate. Nevertheless, in order to face the imminent MPM incidence peak, now more than ever it’s urgent to create a standardized protocol including HITHOC, by joining all our efforts in more exhaustive, large and randomized studies.

## Figures and Tables

**Table 1 jcm-10-03801-t001:** Advantages and limits of HITHOC.

**Advantages of HITHOC**	Multimodal treatment within a single procedure (surgery + chemotherapy + hyperthermia)
	Possibility to avoid demolitive surgery by enhancing cytoreduction with other treatments
	Good tolerability of the procedure by patients with a low morbidity-rate and a rapid post-operative recovery
	Lower systemic toxicity compared to traditional chemotherapy
	Compatibility with all the other adjuvant therapies
**Limits of HITHOC**	Limited indications related to patient performance status, the histology and stage of MPM
	Dedicated equipment and qualified and experienced staff
	Possibility of adverse events such as cardio- and nephrotoxicity, according to the regimens of chemotherapy used
	Prolonged timing of surgical and anesthesiological procedure

**Table 2 jcm-10-03801-t002:** HITHOC protocols used and oncological outcomes. OS = Overall Survival.

Author (Year)	Number of Patients	Drugs (Dosage)	Duration of Perfusion (min)	Temperature (°C)	Survival Rates
Rusch [27] (1994)	28	Cisplastin (100 mg/m^2^) Mitomycin (8 mg/m^2^)	240	ambient	1 year OS rate: 68% 2 years OS rate: 40%
Yellin [28] (2001)	7	Cisplatin (150–200 mg)	60	40.8	1 year OS rate: 72% 2 years OS rate: 65% 3 years OS rate: 44%
Richards [29] (2006)	44	Cisplatin (50–250 mg/m^2^)	60	42	Median OS: 9 months High dosage (175–250 mg/m^2^) Cisplatin: OS 18 months Low dosage (50–150 mg/m^2^) Cisplatin: OS 6 months
Zellos [30] (2009)	29	Cisplatin Amifostine	60	42	Median OS: 17 months High dosage Cisplatin: OS 26 months Low dosage Cisplatin: OS 16 months
Tilleman [31] (2009)	92	Cisplatin (225 mg/m^2^) Amifostine (910 mg/m^2^)	60	42	Median OS: 12.8 months
Sugarbaker [32] (2013)	72	Cisplatin (175–225 mg/m^2^)	60	42	Median OS: 35.3 months
Ambrogi [25] (2018)	49	Cisplatin (80 mg/m^2^) Epirubicin (25 mg/m^2^)	60	42.5	Median OS: 22 months Early stage: OS 35 months Advanced stage: OS 17 months
Burt [33] (2018)	104	Cisplatin (175–225 mg/m^2^) Gemcitabin (100–1200 mg/m^2^)	60	40–42	Median OS: 20.3 months
Klotz [34] (2019)	71	Cisplatin (200 mg/m^2^) Doxorubicin (100 mg/m^2^)	90	42	Median OS: 17.9 months

**Table 3 jcm-10-03801-t003:** Surgical procedures performed and perioperative outcomes. EPP= extrapleural pneumonectomy; P/D = pleurectomy/decortications; PP = partial pleurectomy; EPD = extended pleurectomy/decortication. * Common Terminology Criteria for Adverse Events (CTCAE) v5.0.

Author (Year)	Number of Patients	Surgery Performed + HITHOC	Perioperative Morbidity (%)	Perioperative Mortality (Number of Patients - %)	HITHOC-Related Side-Effects (%)
Rusch [27] (1994)	28	P/D	Not reported	Not reported	Grade 4 * renal toxicity (% not reported)
Yellin [28] (2001)	7	1 EPP 2 P/D 3 PP 1 PP + wedge	bleeding (14.3%) prolonged air leak (14.3%)	1 (14%)	thrombocytopenia (14.3%) nausea (14.3%) non-infectious fever (14.3%)
Richards [29] (2006)	44	EPD	atrial fibrillation (32%) deep-venous thrombosis (9%)	5 (2.2%)	Renal toxicity (75%)
Zellos [30] (2009)	29	EPP	atrial fibrillation (66%) deep venous thrombosis (31%)	2 (0.6%)	Grade 3 * renal toxicity (31%)
Tilleman [31] (2009)	92	EPP	atrial fibrillation (23.9%) deep-venous thrombosis (13%) laryngeal nerve dysfunction (% not reported)	5 (4.6%)	Renal toxicity (9.8%)
Sugarbaker [32] (2013)	72	EPP P/D	Not reported	Not reported	0
Ambrogi [25] (2018)	49	P/D	Prolonged air leak (4%) Grade 3 * anemia (18.4%)	0	0
Burt [33] (2018)	104	EPP P/D Debulking	respiratory distress syndrome (3.8%) atrial fibrillation (4.8%) deep venous thrombosis (2.9%)	2 (1.9%)	Grade 4 renal toxicity (3.8%) Grade 3 leukopenia (1.9%)
Klotz [34] (2019)	71	P/D	prolonged air leak (28.2%) pneumonia (16.9%) respiratory failure (7%) arrhythmia (23.9%)	1 (1.4%)	Renal toxicity (1.4%)

## Data Availability

The data presented in this study are available on request from the corresponding author.

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
