# Peer review of "Hyperthermic Intrathoracic Chemotherapy for Malignant Pleural Mesothelioma: The Forefront of Surgery-Based Multimodality Treatment"

_jcm, 2021, doi:10.3390/jcm10173801_

Round 1
Reviewer 1 Report
Dear authors,
I had the pleasure to examine your overview article on HITHOC. The manuscript is well structured and presents the problems that still exist in the clinical application of HITOC. I agree with the authors that HITHOC needs to be further investigated and should at least be mentioned in international guidelines as an additional local therapy option in the future. I have some suggestions for improvement of your manuscript:
1. Description of possible postoperative renal insufficiency as a frequently mentioned complication after surgical resection and cisplatin-based HITOC. For this, also update the literature: e.g. Markowiak T et al J Surg Oncol. 2019 Dec;120(7):1220-1226.
doi:10.1002/jso.25726. ; Hod T et al. J Thorac Cardiovasc Surg 2021 Apr;161(4):1510-1518. doi: 10.1016/j.jtcvs.2020.05.033. Lapidot M et al. 2020 Dec 3. doi: 10.1097/SLA.0000000000004306. Burt BM 2018 Sep;13(9):1400-1409. doi: 10.1016/j.jtho.2018.04.032.
2. A standardised protocol for HITOC is still missing. Please describe in more detail what factors should be defined here and what experience already exists in the literature. Dosage of cisplatin? Mono oder combination chemotherapy? Duration of HITOC, ...
3. A prospective, randomised study is always called for and would of course be desirable. But why has this not yet been done? What are the problems of such a study?
4. I recommend further additions and updates to the literature, as there are already more papers on clinical analyses and basic research.
I think there are still a few changes and additions to be made to your manuscript before it can be accepted for publication from my point of view.
Reviewer 2 Report
Dear Editor,
I have reviewed the work titled: "Hyperthermic intrathoracic chemotherapy for Malignant Pleural Mesothelioma: the forefront of surgery-based multimodality treatment." by Aprile et al. The study is a review of the combined treatment of malignant mesothelioma, with emphasis on hyperthermic intrathoracic chemotherapy, as a local treatment. In this review, the authors analyze the works published since 1994 to date, giving reference points of the usefulness of local chemotherapy at high temperature. In general, the study is well developed, and the working hypothesis is resolved during the proposed analysis. Below I detail some changes that must be applied in the study:
1) Introduction: Abbreviations that have been defined previously are used. If it is required to use them, they must be indicated in advance.
2) On line 169, the sentence is incomplete.
3) It would be necessary to add a table with additional information on the points in favor and against to use hyperthermic intrathoracic chemotherapy.
Round 2
Reviewer 1 Report
Thank you for the good revision.